# Investigation of Grain, Grain Boundary, and Interface Contributions on the Impedance of Ca$_2$FeO$_5$

**Ram Krishna Hona** [1,*], **Gurjot S. Dhaliwal** [2] and **Rajesh Thapa** [3]

1   Environmental Science Department, United Tribes Technical College, Bismarck, ND 58504, USA
2   Intertribal Research and Resource Center, United Tribes Technical College, Bismarck, ND 58504, USA; gdhaliwal@uttc.edu
3   Feik School of Pharmacy, University of Incarnate Word, San Antonio, TX 78212, USA; thapa@uiwtx.edu
*   Correspondence: rhona@uttc.edu

**Abstract:** Conductivity properties such as the impedance contributions of grain, grain boundary, and electrode–material interface of brownmillerite-type Ca$_2$Fe$_2$O$_5$ are studied using alternate current (AC) impedance at different temperatures over a wide range of frequencies. The compound was synthesized at 1000 °C by a solid-state reaction. Powder X-ray diffraction confirmed the pure and single-phase formation. The correlation of the electrical properties with the microstructure of the compound was studied by an AC impedance spectroscopic technique at different temperatures (25–300 °C), which demonstrated the contribution of both the grain (bulk) and grain boundary to the impedance. The frequency-dependent electrical conductivity was used to study the conductivity mechanism. The electric impedance and the frequency at different temperatures supported the suggested conduction mechanism.

**Keywords:** brownmillerite; impedance; grain; grain boundary and interface

## 1. Introduction

Since brownmillerites have compositional flexibility and are applicable as oxygen ion conductors in solid oxide fuel cells (SOFC) [1], as a membrane for oxygen separation [2], as an electrocatalyst for oxygen or hydrogen evolution reactions, or a catalyst for hydrocarbon oxidation [3,4], there have been intense studies conducted on them. In addition, brownmillerite-type compounds have also been studied for their significant magnetoresistance effects [5,6].

Oxygen defects in the oxygen-deficient perovskites play a considerable role in their properties and can be distributed randomly or in an ordered fashion. Ca$_2$FeO$_5$ is one of the most popular long-range defect-ordered perovskites. Ca$_2$FeO$_5$ has a brownmillerite structure where tetrahedral structures are formed due to defects, and these tetrahedra form alternate layers with octahedra in the structure of a crystal. Due to the long-range defect order, this material has been in focus for oxide ion conductivity and other properties and applications. It is also used as a parent compound to study the doping effect on the structure and properties [7]. Ca$_2$FeO$_5$ has been studied for a wide variety of purposes, such as the production of biodiesel using soybean and Jatropha oils by magnetic CaFe$_2$O$_4$–Ca$_2$Fe$_2$O$_5$-based catalysts [8], ion conductivity in Li-ion batteries [9], and as an electrocatalyst for oxygen evolution reactions [3]. Therefore, Ca$_2$FeO$_5$ is one of the most popular oxygen-deficient perovskites in defect-ordered materials research.

Brownmillerites have also been studied for their electric and magnetic properties due to possible application of their ferroelectricity, piezoelectric, and pyroelectric properties. The electrical properties of brownmillerites were generally studied by measuring the electrical conductivity by the direct current (DC) method or electrochemical impedance method. The DC method is generally used to obtain the information of the total electrical

conductivity at a given temperature, which is the sum of the electronic and ionic conductivities. The DC technique was also found to apply to measuring oxide ion conductivity in brownmillerite-type compounds [10,11]. DC conductivity measurement was applied in $Ca_2FeO_5$ to demonstrate the dependence of charge transport properties on the synthetic technique [12]. We measured the DC conductivity of this material to show the correlation of the electrocatalytic activity of the oxygen evolution reaction with electrical conductivity [3,12]. However, the complete electrical properties of a material cannot be studied by DC measurement. For example, if the conductivity or resistivity contribution of the grains, grain boundaries, and electrode–material interface is to be studied, an electrical impedance spectroscopic (EIS) study is performed. EIS is a powerful tool to measure various electrochemical and electromagnetic properties as well as the charge transport mechanisms. It provides information on the grains, grain boundary, and electrode material interface's contributions to the impedance during charge transport [13,14]. EIS studies can be found to apply during the study of the electrochemical properties of brownmillerites or other materials [15]. It is used to study the magnetoelectric properties as well [16].

It can be found that different spectroscopic techniques have been applied for the study of brownmillerite-type $Ca_2FeO_5$. For example, Mossbauer spectroscopy was used to study the structural relation to the nuclear electric field at the iron site [17] and determination of the nuclear electric field gradient (EFG) tensor parameters at the $Fe^{3+}$ sites in $Ca_2Fe_2O_5$ [18]. Direct current and Seebeck coefficient (thermopower) measurements were performed to study the conductivity [19] and thermoelectric properties [20]. Optical microscopy and energy-dispersive X-ray spectroscopy were utilized to investigate and compare the growth of single crystals and polycrystalline brownmillerite $Ca_2Fe_2O_5$ [21]. The efficiency of the steam–iron process was used for the study of a thermodynamic property of $Ca_2Fe_2O_5$ to produce hydrogen [22]. Many computational studies can also be found for the investigation of this material [23,24]. However, EIS studies, as far as we know, have not been performed yet. This paper reports the EIS study of $Ca_2Fe_2O_5$ for its conductivity mechanism with the frequency-dependent AC impedance to understand its conduction mechanism better.

## 2. Experimental Set-Up

$Ca_2Fe_2O_5$ was synthesized in the air by the solid state reaction method. The precursor compounds were powders such as $CaCO_3$ (99.95% purity) and $Fe_2O_3$ (99.998% purity) purchased from Alfa Aesar (Haverhill, MA, USA), and they were mixed in the proportion of stoichiometry. The mixture was mixed to uniformity with an agate mortar and pestle. The uniformly mixed powder was pressed into a pellet shape with 3 tons of pressure and heated at 1000 °C for 24 h in the air. The pellets were ground and sintered multiple times at 1000 °C for 24 h. For resintering, the powder was again pressed into a pellet 1–2 mm thick and with a diameter of 10 mm using a hydraulic press.

The heating rates were adjusted to 100 °C/h. The purity of the phase and the polycrystalline structure of the samples were confirmed by powder X-ray diffraction (XRD) using Cu K$\alpha$1 radiation ($\lambda$ = 1.54056 Å). Rietveld refinements were performed with the help of GSAS software [16] and an EXPEGUI interface [17]. The microstructure of the samples was studied with a scanning electron microscope (SEM). AC electrochemical impedance spectroscopy (EIS) was used to measure the electrical impedance ($Z$) and capacitance ($Cp$) as a function of the frequency (from 0.1 Hz to 1 MHz) at various temperatures with the help of a computer-controlled frequency response analyzer. The measurement was performed in an open atmosphere in an MTI muffle furnace. For the measurements, the circular pellets were gold-plated on the two sides to collect the current. The gold plating was dried at 200 °C for 30 min to remove the binder. The ramping temperature between different measurements was 3 °C/min. The temperature of the sample was stabilized for 45 min before the impedance measurement. Oxygen deficiency was calculated from iodometric titrations as support for XPS. Iodometric titration was performed to investigate the oxygen stoichiometry and was accomplished by dissolving about 50 mg of a sample and the excess KI (~2 g) in 100 mL of 1 M HCl. Then, 5 mL of the solution was titrated against 0.025 M

$Na_2S_2O_3$. A starch solution of 0.2 mL was added as an indicator to the solution near the endpoint of the titration. All of these steps were accomplished under an argon atmosphere. X-ray photoelectron spectroscopy (XPS) was accomplished with Al K$\alpha$ radiation (1486.7 eV) to investigate the Fe oxidation state.

## 3. Results and Discussion

### 3.1. Material Synthesis and Characterization

$Ca_2Fe_2O_5$ was synthesized by the solid state synthesis technique. The crystal structures of $Ca_2Fe_2O_5$ had been previously reported [3]. XRD, XPS, and SEM were utilized to elucidate the structural properties of the material further. Figure 1 shows the powder XRD data of $Ca_2Fe_2O_5$. The crystal structure of the material was confirmed by Rietveld refinement using the data obtained from its powder X-ray diffraction. Our Rietveld refinement result was consistent with the previously reported structures (brownmillerite) with the *Pnma* space group. Figure 1 shows the alternating arrangement of tetrahedra and octahedra with orthorhombic unit cells. The tetrahedra were oriented in the same direction in the same layer and in the opposite directions in the alternating layers, represented by different colors. The red tetrahedra represent an orientation toward the right, and the blue tetrahedra represent an orientation toward the left. Table 1 lists the refined structural parameters of $Ca_2Fe_2O_5$.

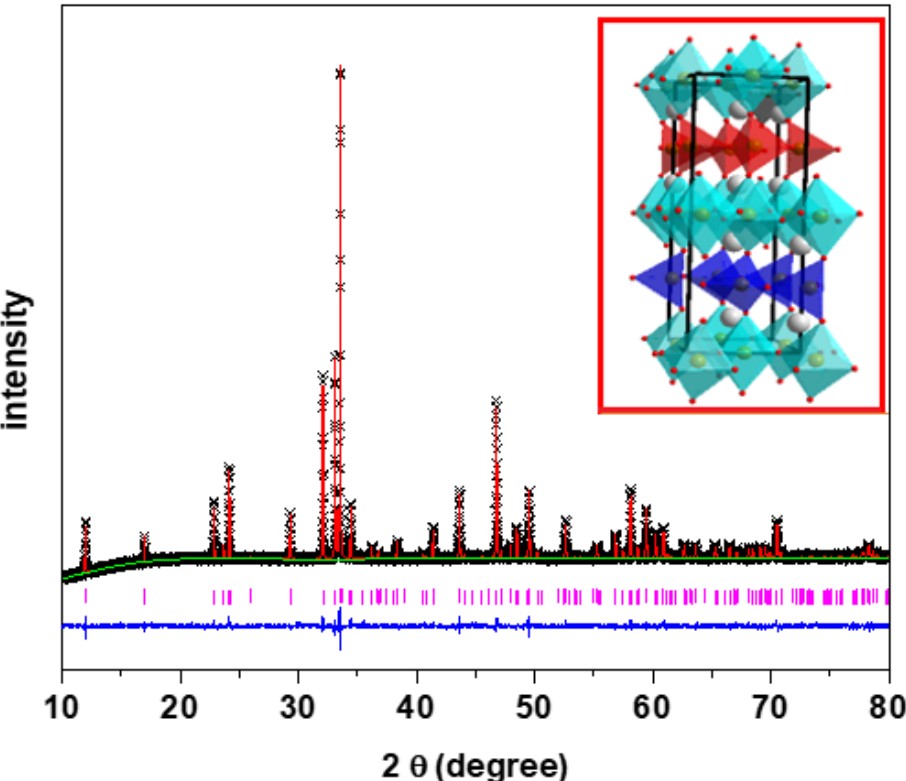

**Figure 1.** XRD data of $Ca_2Fe_2O_5$ with Rietveld refinement. Crosses and vertical pink tick marks represent experimental data and Bragg peak positions, respectively. The solid red line is the *Pnma* model, and the lower blue line is the plot difference.

**Table 1.** The structural parameters for $Ca_2Fe_2O_5$ obtained from the refinement of powder X-ray diffraction. Space group: *Pnma*, *a* = 5.4247 (5), *b* = 14.7756 (1), *c* = 5.5977 (5), *Rp* = 0.013, *wRp* = 0.018, and $\chi 2$ = 1.941.

| Elements | X | y | z | Occupancy | $U_{iso}$ |
|----------|-----|-----|-----|-----------|-----------|
| Ca | 0.4820 | 0.1081 | 0.0226 | 1.0000 | 0.006 (5) |
| Fe1 | 0.0000 | 0.0000 | 0.0000 | 1.0000 | 0.001 (5) |
| Fe2 | −0.0512 | 0.2500 | 0.0680 | 1.0000 | 0.003 (8) |
| O1 | 0.2838 | −0.0141 | 0.2345 | 1.0000 | 0.019 (9) |
| O2 | 0.0315 | 0.1394 | 0.0811 | 1.0000 | 0.009 (4) |
| O3 | 0.5878 | 0.2500 | 0.1248 | 1.0000 | 0.007 (9) |

The scanning electron microscopy image shows close contact between the crystallites of $Ca_2Fe_2O_5$, as shown in Figure 2. As seen in the SEM image, the material was not porous. The non-porosity was also examined gravimetrically. The theoretical calculation was conducted based on the method described in the literature [25]. The experimental density of $Ca_2Fe_2O_5$ was observed to be 98% of the theoretical density. This also illustrates the nonporous nature of the bulk material. Though the crystallites were different in size, the big crystallites dominated the smaller ones. Generally, the $2P_{3/2}$ peaks for trivalent Fe were observed at ∼ 710–711 eV [26,27]. Along with this peak, a satellite peak was observed at about 8 eV higher than the $2P_{3/2}$ peak to confirm the presence of trivalent Fe [26,27]. The XPS data for $Ca_2Fe_2O_5$ (Figure 3) demonstrate the presence of the $2P_{3/2}$ peak observed at ~710 eV, which was followed by a satellite peak observed at ∼ 718 eV, confirming the trivalent state of Fe. The XPS data were further supported by iodometric titrations. The titration showed an oxygen stoichiometry of 5.0 mol per formula unit (i.e., $Ca_2Fe_2O_5$). The matching data of titration and XPS confirmed the oxidation state of Fe to be 3+.

### 3.2. Electrical Properties

The polycrystalline sample-based electrical properties were studied using electrochemical impedance spectroscopy (EIS). EIS is a powerful experimental method that utilizes a small-amplitude alternating current (AC) signal to probe a material's impedance characteristics. This technique measures the impedance of a system over a range of frequencies to generate an impedance spectrum for the electrochemical material under testing. It can give information on the charge transfer mechanisms such as the frequency-dependent impedance, relaxation time, grain, grain boundary, and electrode contribution in a material. For our material, EIS measurements were accomplished in a frequency range from 0.1 Hz to 1 MHz using a frequency response analyzer. The total conductivity was determined from the resistance, which is the intercept of the Nyquist plot at a high frequency.

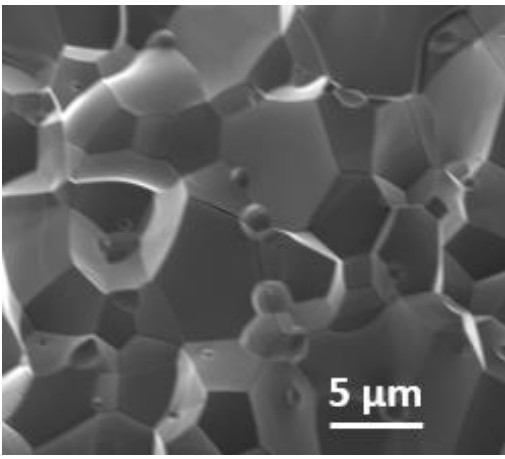

**Figure 2.** SEM image of $Ca_2Fe_2O_5$.

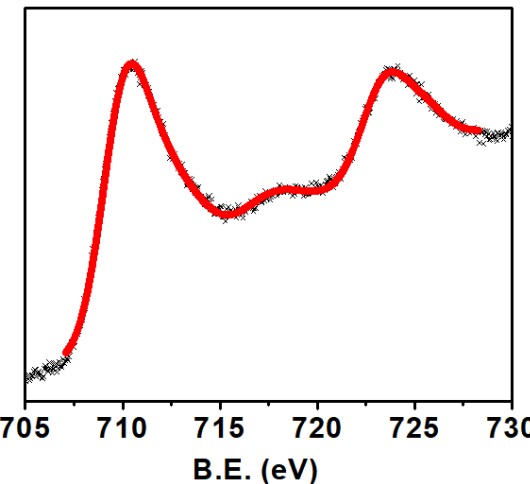

**Figure 3.** XPS spectra of $Ca_2Fe_2O_5$ for Fe.

Figure 4 shows the Nyquist plot for $Ca_2Fe_2O_5$ over a wide frequency range at different temperatures. The EIS data of the material under investigation showed a depressed semicircle where the radius of the semicircle decreased with the temperature rising. If the width of the semicircles decreases with the temperature, the decrease in resistance (or increase in conductivity) with the temperature indicates a semiconductor behavior. The depression was observed for all the semicircles at all temperatures without having centers on the real axis. Such non-Debye-type behavior was considered due to the relaxation time distribution within the bulk material. The ideal EIS of such materials exhibits a semicircle. When such an oxygen-deficient perovskite is measured as an electrolyte, the impedance contribution from the grain, grain boundary, and the electrode–material interface can be considered. Therefore, three semicircular arcs could be expected, corresponding to the grain (bulk), grain boundary, and electrode contributions in the ceramic materials. Three RC elements in a series combination could be used for the best fitted equivalent circuit of the material (Figure 4), where R1, R2, and R3 represent the resistance offered by the grain ($R_g$), grain boundary ($R_{gb}$), and electrode–material interface ($R_{el}$), respectively, and CPE1, CPE2, and CPE3 represent the capacitance of the grain, grain boundary, and electrode–material interface, respectively. A constant phase element (CPE) is generally applied for the nonideal capacitive behavior demonstrated by an imperfect semicircle. The nonideal behavior is caused by surface roughness, leakage capacitance, and nonuniform distribution [28]. In our material (Figure 4), the heights of the semicircles were observed to be smaller than half of their diameter. Therefore, the classical parallel R//C circuit used for a perfect semicircle was not the suitable equivalent circuit to model this type of data of depressed semicircles [26,28–34]. The inset demonstrates the equivalent circuit used, where the real (Z′) and imaginary (Z″) part of the impedance were derived by Equations (1) and (2) [30,31].

$$Z' = \frac{R_g}{1 + \left(R_g C_g\right)^2} + \frac{R_{gb}}{1 + \left(R_{gb}C_{gb}\right)^2} + \frac{R_{el}}{1 + (R_{el}C_{el})^2} \tag{1}$$

$$Z'' = \frac{\omega R_b^2 C_b}{\left[1 + (\omega R_g C_g)^2\right]} + \frac{\omega R_{gb}^2 C_{gb}}{\left[1 + \left(\omega R_{gb}C_{gb}\right)^2\right]} + \frac{\omega R_{el}^2 C_{el}}{\left[1 + (\omega R_{el}C_{el})^2\right]} \tag{2}$$

where $\omega$ is the angular frequency. The electrical behavior of the semiconducting material can be explained by the semicircles and their points of intercept on the real (Z′) axis. These intercepts give the values of the bulk ($R_g$), grain boundaries ($R_{gb}$), and material–electrode interface ($R_{el}$).

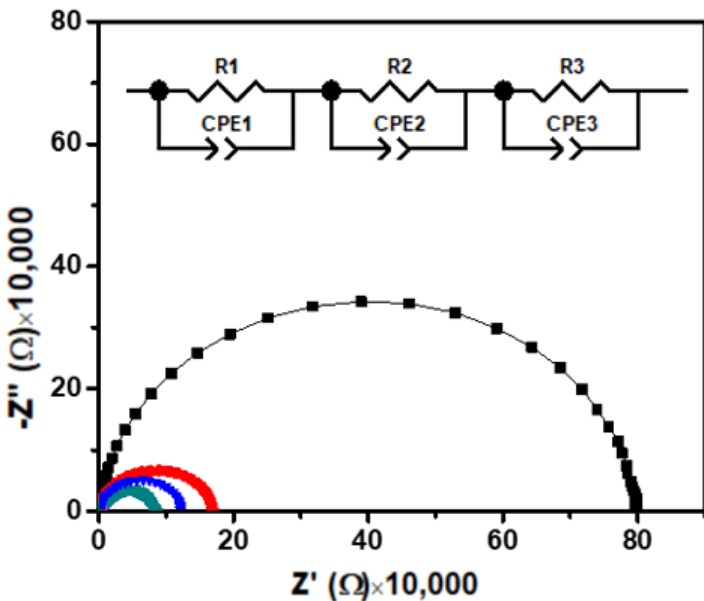

**Figure 4.** Nyquist plots of $Ca_2Fe_2O_5$ at different temperatures. Inset shows the equivalent circuit used to fit the data, where R1, R2, and R3 are contributions due to grain ($R_g$), grain boundary ($R_{gb}$), and interface ($R_{el}$), respectively. Similarly, CPE1, CPE2, and CPE3 represent the capacitance due to grain ($C_g$), boundary ($C_{gb}$), and interface ($C_{el}$), respectively. (Black, red, blue, and green represent the measurements at 25, 100, 150, and 200 °C, respectively. The data measured at 250 and 300 °C are not seen due to overlap near the origin with lower temperature data.)

Figure 5 helps to illustrate the frequency-dependent Z′ and Z″ fits, which correspond to the proposed equivalent circuit. These spectra assist in validating good agreement between the theoretical and experimental data and in elucidating the electrical properties of the sample through the proposed electrical circuit [32].

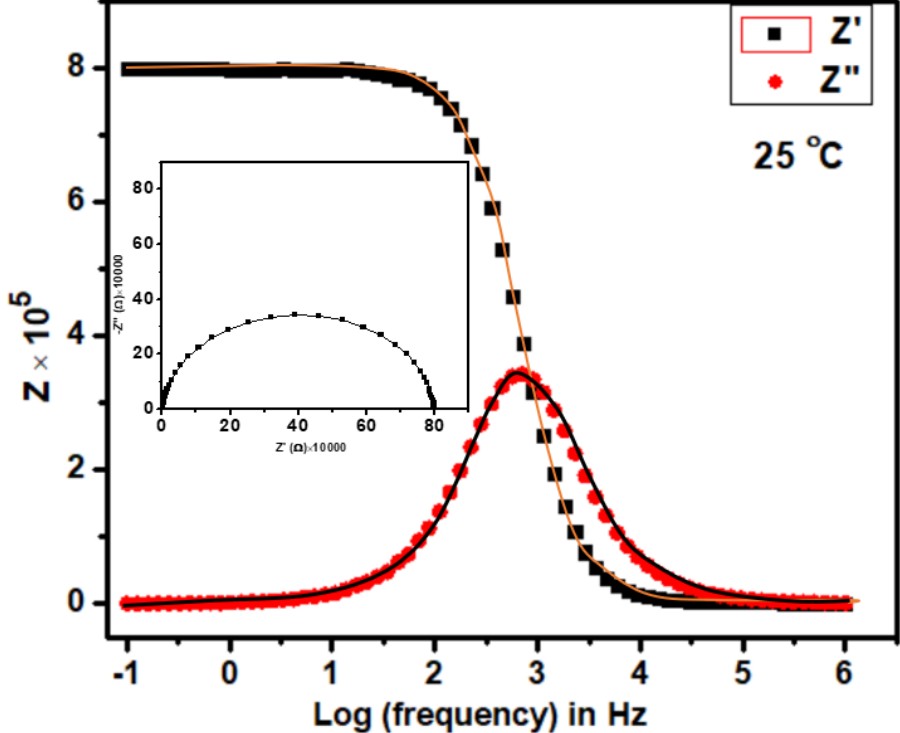

**Figure 5.** Variation of Z′ and Z″ with frequency at 25 °C. Inset is the cole-cole plot at the same temperature for the compound $Ca_2Fe_2O_5$.

The intercept of the semicircular curve on the low-frequency region in Figure 4 gives the resistance of the material under study. This resistance (R) provides the conductivity with σ of the bulk material (circular pellet) of the thickness (L) and surface area (A) according to Equation (3) [33]:

$$\sigma = L/(RA) \tag{3}$$

The total resistance $R_{total}$ is calculated using the intercept of the semicircle on the low-frequency region of the real axis (Z′), which is equivalent to the total of the individual contributions such that

$$R_g + R_{gb} + R_{el} = R_{total} \tag{4}$$

Table 2 displays the values of $R_g$, $R_{gb}$,, and $R_{el}$ for the material $Ca_2Fe_2O_5$ at different temperatures. Figure 6 clearly shows how the grain, grain boundary, and interface contributions were affected by the temperature variation. As seen in Figure 6, the grain and electrode–material interface showed similar patterns of temperature effects, where the contributions first increased from 25 °C to 100 °C and then decreased until 300 °C. However, the grain boundary had a different temperature effect where the contribution first reduced from 25 °C to 100 °C and then very slowly increased until 300 °C. It seems that the contributions were almost similar at 300 °C for $R_g$, $R_{gb}$, and $R_{el}$. The values were almost identical at 300 °C, as seen in Table 2.

**Table 2.** Grain ($R_g$), grain boundary ($R_{gb}$), and electrode–material interface ($R_{el}$) contributions at different temperatures.

| T (°C) | $R_g$ (Ω) | $R_{gb}$ (Ω) | $R_{el}$ (Ω) |
|--------|-----------|--------------|--------------|
| 25 | 2364 | 605,910 | 190,730 |
| 100 | 217,070 | 2443 | 568,380 |
| 200 | 13,771 | 203 | 73,776 |
| 300 | 14.22 | 13.26 | 4.48 |

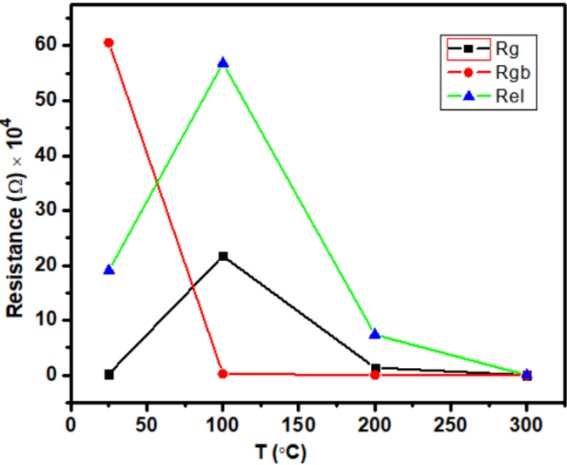

**Figure 6.** Variation of $R_g$, $R_{gb}$, and $R_{el}$ depending on the temperature for the compound $Ca_2Fe_2O_5$.

The frequency and temperature-dependent imaginary part of the impedance (Z″) for $Ca_2Fe_2O_5$ are shown in Figure 7. Each curve demonstrates a peak (at the Z″ max value) at a frequency, and this peak position is conventionally known as the relaxation frequency ($f_R$). Hence, our material showed the existence of relaxation phenomena due to the presence of the peaks. A single peak represents a single relaxation process. Our material also showed a single peak at lower temperatures. If a peak is narrow, it represents a narrow relaxation time distribution. Our material demonstrated a relatively narrow peak at 25 °C, indicating a narrow distribution of relaxation times at this temperature.

The relaxation peaks were due to the presence of immobile species or electrons at low temperatures and defect vacancies at high temperatures [32]. The progressive widening of this peak with the increase in temperature revealed a gradual widening of the distribution, suggesting the various electrical processes during the distribution of relaxation times [34]. From Figure 6, we can observe that the $Z''$ max values decreased with the increase in temperature, indicating a semiconductor behavior of $Ca_2Fe_2O_5$. Except at 25 °C, the peaks in $Z''$ were found to shift toward a higher frequency as the temperature decreased. This result is opposite to those of other reports, which mentioned a decrease in the relaxation time [35]. The shifting of the peaks with the temperature variation was considered to be due to electrical processes such as conductivities and relaxations at the peak frequencies [36].

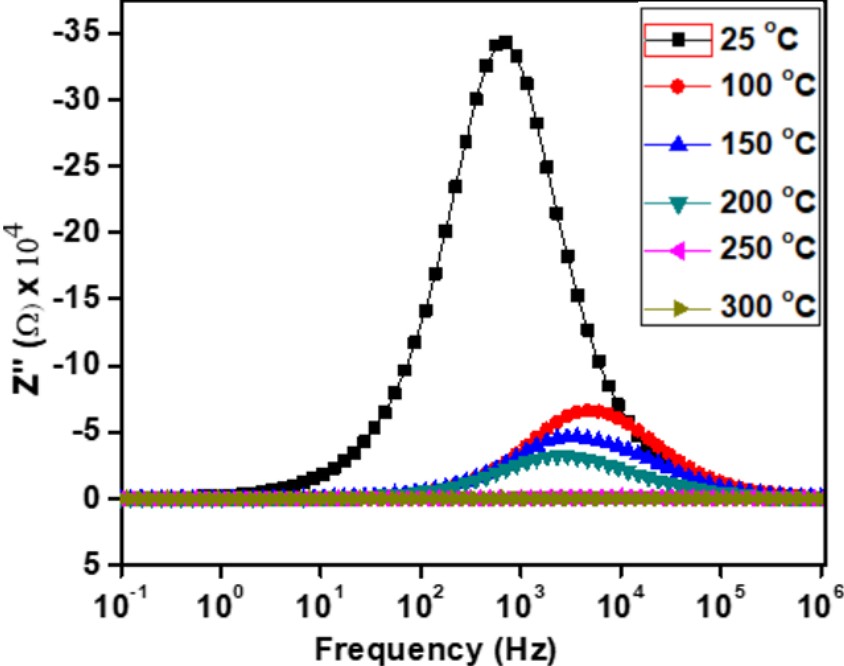

**Figure 7.** Frequency-dependent imaginary part of impedance ($Z''$) at various temperatures.

Figure 8 shows the variation of the real part ($Z'$) of impedance with the frequency at the temperature range of 25–300 °C for $Ca_2Fe_2O_5$. It can be clearly observed from Figure 8 that the $Z'$ value in the low-frequency region was higher at lower temperatures and decreased gradually with an increasing temperature, indicating an increase in the AC conductivity with the increase in temperature. This displays the semiconductive behavior of the material as described in Figure 4. This type of behavior has been reported in several other perovskite oxides and has been regarded as a result of reducing the grains, grain boundaries, and electrode–interface resistance. This is a result of the negative temperature coefficient of the sample resistance [37]. The $Z'$ values started sloping down at a certain frequency (near 1 kHz), and the sloping continued for all temperatures with the increase in frequency. This type of result (i.e., lower values of $Z'$ at higher frequencies) has been seen in many perovskite oxides and has been regarded as a result of the release of space charge polarization. This phenomenon has been considered to be responsible for the enhancement of conductance at high frequencies. As the $Z'$ curve shifts toward a higher frequency, the $Z'$ curves of all the temperatures start merging [36,37]. The merger of the $Z'$ curves at the higher frequency region is considered to be due to the release of space charges or polarization and due to the reduction in the material's barrier properties with an increasing temperature. It can also be interpreted by the presence of space charge polarization [36,37]. The positions which were frequency independent in the $Z'$ curves were found to shift toward the higher frequency side at higher temperatures. This type of phenomenon has

already been reported in several other perovskite oxides [32,38]. It is considered to be due to the frequency relaxation process in the material [32].

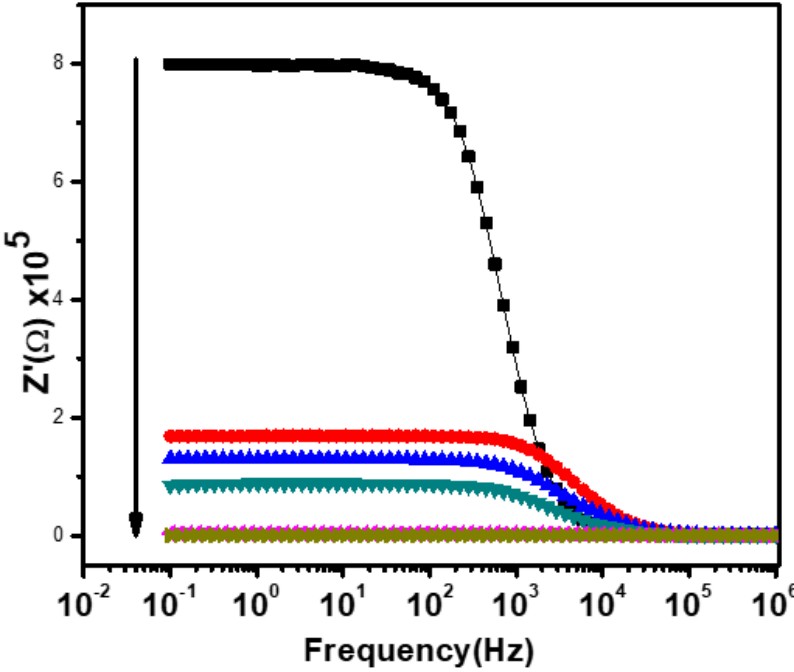

**Figure 8.** The frequency-dependent real part impedance at various temperatures. Black, red, blue, green, pink, and brown represent the measurements at 25, 100, 150, 200, 250, and 300 °C, respectively.

### 4. Conclusions

The frequency-dependent AC conductivity of $Ca_2Fe_2O_5$ synthesized by a solid state reaction was investigated at various temperatures ranging from 25 to 300 °C and frequencies ranging from 0.1 Hz to 1 MHz. The EIS data suggest that the relaxation mechanism of $Ca_2Fe_2O_5$ was observed with the mixed effect with varying temperatures. It is also shown that the grain resistance ($R_b$), grain boundary ($R_{gb}$), and interface resistance ($R_{el}$) varied with increasing temperatures, and the grain boundary effect was dominant at 25 °C. The interface effect was dominant at 100 °C and 200 °C. However, the contribution at 300 °C seemed to be almost equal for all 3 components.

**Author Contributions:** Data collection and analysis, R.K.H. Funding acquisition and resources, G.S.D. Conceptualization, R.T. All authors have read and agreed to the published version of the manuscript.

**Funding:** The co-author Gurjot S. Dhaliwal was supported for this research by the National Science Foundation (HRD 1839895).

**Institutional Review Board Statement:** Not applicable.

**Informed Consent Statement:** Not applicable.

**Data Availability Statement:** Data is contained within the article.

**Conflicts of Interest:** The authors declare no conflict of interest.

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
