# Peer review of "Investigation of Grain, Grain Boundary, and Interface Contributions on the Impedance of Ca2FeO5"

_applsci, doi:10.3390/app12062930_

Round 1
Reviewer 1 Report
Please find my considerations on the manuscript in the attached file

Author Response
We have attached the reply to the reviewers

Reviewer 2 Report
In this work, electrical properties of oxygen-deficient CaFe2O5 were studied experimentally by EIS. For this purpose, the samples were made by a solid state reaction method. Their sintered forms were well analyzed by several methods. The results indicated that the samples were good enough. The impedance measurements show that:
_ the total resistance measured were well decomposed into Rg, Rgb, Rel; at 25 degrees C, Rg is relatively low compared to Rgb, but this feature is reversed at 100 and 200 degrees C.
_ Z’ and Z” each shows it’s variation with respect to A.C. frequency and temperature, clearly attributed to electrons, defect vacancies, space charges, polarizations.
The paper is readable, but the following points must be considered for much improvement:
_ several typos are seen over the manuscript, and must be corrected.
_ some expressions are difficult to understand (the problem may be logical sequences of sentences), which can be improved with the help of professional English revision service.
_ it is recommended that an simple equation to obtain Rg, Rgb, Rel is written in the text.
_ it is recommended that area A in equation (1) is written as L/(RA), for no confusion.
_ it is hard to see widths of curves in Figure 5, which can be improved by modifying the graphical expression, e.g. putting an inset in the figure or illustrating.
_ Z” behavior with respect to frequency in Figure 5 must be explained in more detail by a way, e.g. illustration, and with respect to temperature also be explained in the same way.
_ curves of Figure6 must be labeled with temperature.
_ Z’ behavior with respect to frequency in Figure 6 must be explained in more detail by a way, e.g. illustration, and with respect to temperature also.
_ in line 213, “interface resistance” is confusing, so it must be written as “this interface resistance” or Rgb.
_ in 216, “polaron” appears suddenly, unexpected, and so sufficient explanation about it must be done in Results and discussion.
Author Response

(The authors gave the same response as above.)

Round 2
Reviewer 1 Report
The authors have thoroughly addressed all my comments therefore the manuscript can be accepted in its present form.
Formatting of Fig.2, Fig.3 and Tab.1 seems a bit odd, maybe is only a problem of the pdf output.
Reviewer 2 Report
The manuscript has been revised and pretty much improved. I accept the current form.